# Extraction, Isolation, Screening, and Preliminary Characterization of Polysaccharides with Anti–Oxidant Activities from *Oudemansiella raphanipies*

**DOI:** 10.3390/polym15132917

**Published:** 2023-06-30

**Authors:** Junqiang Qiu, Wang Shi, Jingnan Miao, Hui Hu, Yanan Gao

**Affiliations:** 1Key Laboratory of Ministry of Education for Advanced Materials in Tropical Island Resources, Hainan University, Haikou 570228, China; 2Key Laboratory of Tropical Translational Medicine of Ministry of Education, School of Pharmacy, Hainan Medical University, Haikou 570100, China; 3School of Chemistry and Chemical Engineering, Harbin Institute of Technology, Harbin 150010, China

**Keywords:** *Oudemansiella raphanipies* polysaccharides, optimization, characterization, NMR, antioxidant

## Abstract

Response surface methodology (RSM) was used to find the optimal extraction process of *Oudemansiella raphanipies* polysaccharides (ORPs). The results showed that the optimal extraction parameters were an alkali concentration of 0.02 mol/L, a ratio of material to liquid of 1:112.7 g/mL, an extraction temperature of 66.0 °C, and an extraction time of 4.0 h. Under the optimal conditions, the yield of ORPs was raised to 16.2 ± 0.1%. The antioxidant activities of ORPs–I~V were determined and compared, and ORPs–V was further purified by chromatography, with an average molecular weight (*Mw*) of 18.86 kDa. The structure of ORPs–V was determined by Fourier transform–infrared spectroscopy (FT–IR), monosaccharide analysis, and nuclear magnetic resonance (NMR) spectroscopy. The ORPs–V comprised fucose, rhamnose, arabinose, glucose, galactose, mannose, xylose, fructose, galacturonic acid, and glucuronic acid at a ratio of 1.73:1.20:1.13:2.87:8.71:2.89:1.42:0.81. Compared to other ORPs, ORPs–V showed the strongest antioxidant activities (ABTS radical cation, hydroxyl radical and DPPH scavenging activities, and reducing power), and were able to significantly increase the activities of superoxide dismutase, catalase, lactate dehydrogenase, and glutathione peroxidase. However, they reduced the malondialdehyde content in mice fed a high-fat diet. These results indicate that ORPs–V may be good anti–oxidant agents to be applied in functional foods.

## 1. Introduction

*Oudemansiella raphanipies* (*O. raphanipies*) has been described uniformly as either *O. radicata* or *O. furfuracea*, and, until recently, was identified as *O. raphanipies* on the basis of findings from morphological characteristics and molecular phylogenetic studies [1]. As an edible and medicinal mushroom, described in India, Australia, Japan, Korea, Thailand, and China successively, *O. raphanipies* has been widely and artificially cultivated in the Hainan, Yunnan, Guangxi, Fujian, and Sichuan provinces of China for its delicious flavor and potential medical value. *O. raphanipies* is called the “Edible Queen” because it contains abundant active components, including polysaccharides, mucidin, oudemansin, oudenone, alkaloid, amino acids, and trace elements [2,3]. Among them, functional polysaccharides from mushrooms have garnered more and more attention because of their excellent development value in foods, cosmetics, medical materials, and drugs [4,5]. *O. raphanipies* polysaccharides (ORPs) have been found to possess various bioactivities, such as anti–oxidation [2], anti–tumor [6], regulation of the favorable intestinal microflora [7], anti–fungal [8], anti–inflammatory [9], hepato–protective, and lung–protective activities [10]. However, as the main active ingredients in *O. raphanipies*, ORPs have seldom been reported in the past decade. Although solvent extraction is regarded as the most important and straightforward strategy for separating biologically active substances, different extraction processing methods exhibited major influences on extraction yield, molecular weight, monosaccharide composition, and functional group [11]. Several studies have shown that alkaline extraction could be considered a practical and economical method with which to obtain polysaccharides with superior bioactivities [12]. It is widely applied to enhance the dissolution of natural polysaccharides in the process of extraction, mainly by destroying the structure of the plant or mushroom cell wall in order to improve the yield of polysaccharides [13,14,15]. 

To our knowledge, there has only been a small amount of research that has separated ORPs from *O. raphanipies* in recent years. The extraction rate, chemical composition, and biological activities of *O. raphanipies* polysaccharides extracted from hot water have each been shown in many studies. Recently, a purified ORP with an average molecular weight (*Mw*) of 40.9 kDa was reported, which showed strong anti–oxidant and anti–inflammatory activities. The ORPs consisted of →6)–*α*–D–Man*p*–(1→4)–*β*–D–Xyl*p*–(1→6)–*α*–D–Gal*p*–(1→[6)–*β*–D–Glu*p*–(1]_4_→ without a branched structure [16]. In addition, a homogeneous ORP (named ORPS–1) comprising galactose, fucose, glucose, mannose, and xylose at a ratio of 18:6:6:4:1 was obtained by DEAE–52 and Sephadex G–200 columns, which was able to significantly improve liver function, ameliorate liver steatosis, and reduce lipid droplet accumulation in mice with high–fat–diet (HFD)–induced nonalcoholic fatty liver disease [17]. Moreover, a homogeneous ORP (named ORP–1) with an average *Mw* of 24.0 kDa was reported, which comprised mannose, glucose, galactose, and fucose at a ratio of 1.48:1.00:29.65:8.22. The ORP–1 could restore a healthy gut microbial population to ameliorate age–related gut microbiota dysbiosis by facilitating the proliferation and adhesion of the probiotics *Lactobacillus acidophilus* and *Bifidobacterium bifidum* [18]. Compared to other mushroom polysaccharides, there is very little available information about the alkaline water extraction, detailed structural elucidation, and antioxidant activities of ORPs. Hence, it is purposeful for us to offer meaningful research findings regarding *O. raphanipies* polysaccharides extracted from alkaline water.

In this study, the response surface methodology (RSM), on the basis of a Box–Behnken design (BBD), was utilized to obtain the optimal extraction processing parameters (alkali concentration, ratio of water to raw material water content, extraction temperature, and extraction time) for ORPs. Then, the crude ORPs were obtained using DEAE–52 and Sephadex G–150 chromatographic columns. The preliminary structural characterization and antioxidant activities of the ORPs were studied using Fourier transform infrared (FT–IR) and gas chromatography–mass spectrometry (GC–MS). Moreover, the in vitro antioxidant activities of the ORPs were investigated using ABTS (2, 2′–azinobis–(3–ethylbenzthiazoline–6–sulphonate)), DPPH (2, 2–diphenyl–1–picrylhydrazyl), reducing power, and a hydroxyl radical scavenging assay. Furthermore, the in vivo antioxidant activities of ORPs–V were evaluated at the same time. 

## 2. Materials and Methods

### 2.1. Materials and Chemicals

The fruit bodies of *O. raphanipies* were purchased from a market base in Haikou City, Hainan Province, China (October 2019). They were washed, vacuum–dried at 60 °C, and crushed into fine particles using a pulverizer (FW177, Tianjin, China) to obtain *O. raphanipies* powder. DEAE–52 cellulose and Sephadex G–150 (2.5 cm × 36 cm) were obtained from Solarbio Technology Co., Ltd. (Beijing, China). Standard monosaccharides (fucose, rhamnose, arabinose, glucose, galactose, fructose, mannose, xylose, galacturonic acid, and glucuronic acid) were purchased from Sigma (St. Louis, MO, USA). DPPH and ABTS were purchased from Sigma–Aldrich (St. Louis, MO, USA). Purified water was used throughout the experiments, and was purified by a Millipore’s Mill–Q water purification system (Bedford, MA, USA). The superoxide dismutase (SOD, A001–3–2), lactate dehydrogenase (LDH, A020–2–2), catalase (CAT, A007–1–1), malondialdehyde (MDA, A003–1–2), and glutathione peroxidase (GSH-Px, A005–1–2) were purchased from the Nanjing Jiancheng Institute of Biological Engineering (Nanjing, Jiangsu, China). All other chemical reagents were of analytical grade and were obtained from local suppliers.

### 2.2. Preparation of Crude ORPs

The *O. raphanipies* powders (100.0 g) were successively extracted with a designed extraction alkali concentration (*X*_1_: 0.01–0.03 mol/L), ratio of water to raw material (*X*_2_: 1: 80.0–: 120.0 g/mL), extraction temperature (*X*_3_: 60.0–80.0 °C), and extraction time (*X*_4_: 3.0–5.0 h). The pH of each aqueous alkali extract was adjusted to neutral using the correct amount of HCl (0.1 mol/L). The NaCl was removed by dialysis (3500 Da, 24 h), and excess ethyl alcohol (ratio of 3:1, *v*/*v*) was added to the solutions to obtain a precipitate. The precipitate was centrifuged at 3000 g (TG16–WS, Xiangyi Centrifuge Instrument Co., Ltd., Changsha, China) for 20 min at 25 °C, then lyophilized (Vaco2, Zirbus, Duesseldorf, Germany). Hence, the crude ORPs were obtained and kept at 4 °C. 

### 2.3. Experimental Design of Animals

A total of 40 male C57BL/6 mice (6 weeks old, weighing 15–20 g) were purchased from Hunan Shrek Jingda Experimental Animal Co., Ltd. (Changsha, Hunan, China; production license number: SCXK [Xiang] 2021–0002), and kept in cages at 25 ± 2 °C and 50 ± 2% humidity with a 12 h light–dark cycle, natural food, and water. After being kept for 7 days, the mice were randomized into either a normal control (NC group, *n* = 8) or a high-fat diet (*n* = 32) group. The mice in the NC group were fed with a normal diet (the total energy was 3.8 kcal/g, including 70% carbohydrates, 20% protein, and 10% fat), whereas the HFD group was fed with a high–fat diet (the diet contained 20% carbohydrates, 20% protein, and 60% fat, and the total energy was 5.2 kcal/g). After 4 weeks of feeding, 32 HFD mice were divided into the following groups at random: HFD (HFD group, *n* = 8); 50 mg/kg ORPs–V (ORPs–V–L group, *n* = 8); 100 mg/kg ORPs-V (ORPs–V–M group, *n* = 8); and 200 mg/kg ORPs–V (ORPs–V–H group, *n* = 8). The mice in the NC and HFD groups were fed with deionized water, whereas mice in the ORPs–V groups were fed with the corresponding amount of ORPs–V. 

After feeding for 12 weeks, all experimental mice were fasted for 12 h, then anesthetized with 1% sodium pentobarbital (25 mg/kg). Subsequently, the blood of the mice was taken through the canthus. After being kept for 60 min, the serum was obtained by centrifugation (3000× *g* for 15 min, 4 °C)(H2050R, Xiangyi Centrifuge Instruments Co., Ltd., Changsha, China). The serums were kept at −80 °C (Zhongke Meiling Cryogenic Technology Co., Ltd., Hefei, China). Management of the mice conformed to the rules and regulations of the Animal Experimental Ethics Committee of Hainan Medical University.

### 2.4. Experimental Design of RSM

The experiment was conducted according to the BBD for the RSM study. The response surface designs required 3 levels, set as −1, 0, and +1, which were applied to evaluate the optimal extraction conditions for ORPs. A BBD with 4 factors and 3 levels was applied to obtain various extraction parameters for the optimization extraction. The design was applied to the quadratic response surface, and we obtained a second–order polynomial regression model. Basis on the single–factor test data (Figure 1), the alkali concentration (*X*_1_), ratio of material to liquid (*X*_2_), temperature (*X*_3_), and time (*X*_4_) were chosen as the independent variables, and their acceptable ranges were assessed as suitable levels for subsequent experimental research. At the central point, the integral design in this study contained 29 combinations with 5 replicates, which are shown in Table 1. All of the combination tests were validated at least 3 times.

The model fitting results and “value of response” (the ORPs extraction rate) at different experimental parameters, including their interactions, are indicated in Table 1. The influence of extraction conditions (alkali concentration, ratio of material to liquid, extraction temperature, and extraction time) on the extraction rate was studied using the Box–Behnken design, which contains 4 factors and 3 levels. There were 5 replicates at the central point. The stability and inherent variability of the extraction experiment were validated by applying 5 center–point runs.

### 2.5. Purification of ORPs

The crude ORPs were purified using the method described by Sun et al. [11]. Briefly, the crude ORPs solution was obtained using distilled water and filtered with a 0.44 μm filter membrane. Then, 10.0 mL of solutions were purified using a column of DEAE–52 (2.9 cm × 60.0 cm), followed by stepwise elution using deionized water and different concentrations of NaCl solutions (0.05, 0.10, 0.15, and 0.20 mol/L) at a stable flow rate of 0.25 BV/h. Eluate (5.0 mL/tube) was obtained successively, the samples were detected using the phenol–sulfuric acid method, and glucose was used as the standard. The main fractions were obtained and dialyzed with deionized water; then, Sephadex G–150 was applied for further purification. The partially purified ORPs (16.0 mg) were dissolved into distilled water (4.5 mL) and placed onto the Sephadex G–150 column (2.5 cm × 36.0 cm). The column was eluted with distilled water at a stable flow rate of 0.32 BV/h, and the main fractions were obtained (6.0 mL/tube) using an automatic fraction collector. The phenol–sulfuric acid method was adopted to determine the polysaccharide content. At last, the main fractions were collected, concentrated, dialyzed with deionized water (3500 Da, 48 h), and lyophilized.

### 2.6. Characterization of ORPs

#### 2.6.1. Fourier Transform Infrared (FT–IR) Analysis

The FT–IR spectra of ORPs were acquired using a TENSOR27 FI–IR spectrometer (Bruker, Pittcon, Germany). The dried ORP samples (5.0 mg) were placed into potassium bromide (KBr, spectroscopic grade) (50.0 mg) and ground into powder. Then, the powders were pressed into pellets 1 mm in diameter at a frequency range of 4000~500 cm^−1^, and the FT–IR spectra of the ORPs were continuously scanned 10 times [19,20]. The FT–IR profile was drawn using Origin Pro 9.0 software.

#### 2.6.2. Determination of Monosaccharide Composition

The purified ORPs (5.0 mg) were hydrolyzed by 3.0 mol/L trifluoroacetic acid, which was kept at 110 °C for 9 h. Superfluous trifluoroacetic acids were eliminated under reduced pressure (≥0.095 MPa) at 35 °C. The hydrolysates were modified into acetylate aldononitrile derivatives. Then, an Agilent 6890–5973 GC (Agilent Company, Santa Clara, CA, USA), equipped with a capillary column (30.0 m × 0.32 mm), was used to measure the samples [21]. Similarly, the monosaccharide standards were prepared and detected by the above–mentioned method. 

#### 2.6.3. Determination of Molecular Weight

The concentration of the ORPs–V solution was kept at 2.0 mg/mL and treated using a 0.45 μm membrane (Tianjin Jinteng Experimental Equipment Co., Ltd., Tianjin, China), then injected into a high–performance gel permeation chromatographer (HPGPC) (Waters 2424, Waters, Milford, MA, USA). The chromatography system consisted of an ultra–hydrogel linear gel filtration column (300.0 mm × 7.8 mm, G–3000 PWXL, Tosoh Co., Ltd., Tokyo, Japan) and a refractive index detector (RID). The elution was obtained with distilled water, and the calibration curve was built on the basis of dextran standards (13,050, 36,800, 64,650, 135,350, 300,600, and 2,000,000 Da) to determine the *Mw* [22]. Empower software (Waters Crop., Milford, MA, USA) was used to process the above results.

#### 2.6.4. Nuclear Magnetic Resonance (NMR) Analysis

A 50.0 mg sample of ORPs–V was dissolved in 500 μL D_2_O for ^1^H analysis and ^13^C analysis. The sample was transferred into a 5 mm NMR tube and assessed using a Bruker DRX–600 MHz NMR spectrometer (NMR Bruker 600 MHz, Rheinstetten, Germany). The ^1^H–^13^C heteronuclear single quantum coherence spectroscopic (HSQC) NMR spectra of ORPs were determined using a Bruker MSI–300 NMR spectrometer (which was tested at 25 °C). The scanning number of the HSQC spectra was 1024. The results were processed using standard Bruker Topspin–NMR software.

### 2.7. Anti–Oxidant Activities of ORPs

#### 2.7.1. Anti–Oxidant Activity Assays In Vitro

The ABTS radical cation (ABTS^+^) scavenging activity was tested according to the method of Yuan et al., with a certain modification [23]. An aliquot of 160 μL of the suitable ABTS^·+^ solution was blended with 40 μL of ORPs at particular concentrations, or a blank control (deionized water) was used, at 25 °C. Ascorbic acid (Vc) was set as the positive control. 15 min later, the final reaction solutions were determined at 734 nm, and the radical scavenging activities were evaluated as:Scavenging activity (%) = (1 − A_sample_/A_blank control_) × 100(1)
where A_sample_ is the absorbance of ORPs and A_blank control_ is the absorbance of the blank control.

The hydroxyl radical–scavenging activities were evaluated according to the method of Wang et al. [24]. To be brief, 9 mmol/L alcohol–salicylic acid and an aliquot of 140 μL of 9 mmol/L FeSO_4_ were blended in a colorimetric tube. After mixing, 60 μL of different concentrations of ORPs or deionized water as a blank control were transferred into the above solutions, respectively. Ascorbic acid (Vc) was set as the positive control. Then, 140 μL of 8.8 mmol/L hydrogen peroxide was blended with the above solutions. The mixed solutions were kept at 25 °C for 20 min. The absorbance of the final solutions was determined at 510 nm, and the hydroxyl radical-scavenging activities were evaluated as:Scavenging activity (%) = 1 − (A_sample_ − A_blank control_)/A_control_ × 100(2)
where A_sample_ is the absorbance of polysaccharides, A_blank control_ is the absorbance of the control without hydrogen peroxide, and A _control_ is the absorbance of the control.

The scavenging ability of ORPs for DPPH free radicals was evaluated according to the previous method [25]. Briefly, ORPs were dissolved in deionized water to obtain ORPs solutions of different concentrations, and 35 μL samples of the ORPs solutions were blended into 165 μL DPPH ethanol solution (0.4 mmol/L) in the dark for 15 min. Ascorbic acid (Vc) was set as the positive control group. The absorbance of the mixed solutions was determined at 517 nm. The hydroxyl radical-scavenging activities were evaluated as:Scavenging activity (%) = 1 − A_sample_/A_blank control_ × 100(3)
where A_sample_ is the absorbance of the mixed solutions of ORPs and DPPH, and A_blank control_ is the absorbance of mixed solutions of deionized water and DPPH. 

The reducing power of ORPs was analyzed according to the method reported by Chen et al. [26]. In brief, aliquots of 35 μL of ORPs solution at different concentrations were blended into 90 μL PBS solutions (0.2 mol/L, pH 6.6) and 90 μL of potassium ferricyanide [K_3_Fe(CN)_6_] (1% *w*/*v* in deionized water), respectively. The mixed solutions were maintained at 37 °C for 25 min, and the above solutions were added into 500 μL of trichloroacetic acid (10% *w*/*v* in water) and centrifugated at 3000× *g* (TG16G, Kaida Scientific Instruments, Changsha, China) for 15 min at 25 °C. The supernatants (500 μL) were blended with 500 μL of deionized water and 0.5 mL of FeCl_3_ (0.1% *w*/*v* in water), and then the absorbance was determined at 700 nm against the blank control group, which consisted of all reagents except the ORPs.
Reducing power = A_sample_ − A_blank control_(4)
where A_sample_ is the absorbance of the polysaccharides and A_blank control_ is the absorbance of the negative control group.

#### 2.7.2. Anti–Oxidant Activity Assays In Vivo

The anti–oxidant activities of ORPs–V in vivo were evaluated according to the method of Zeng et al. [27]. Briefly, an enzyme–labeling instrument (MR–96A, Mindray Instruments, Inc., Shenzhen, China) was used to measure the activities of antioxidant enzymes (CAT, SOD, LDH, and GSH–Px) and MDA contents in the mice serum according to the manufacturer’s instructions.

### 2.8. Statistical Analysis

All operations and experiments were conducted 3 times, and all of the data are shown herein as mean ± SD. The statistical analysis results of the single–factor data were obtained by Origin 9.0 (OriginLab Corp., Northampton, MA, USA) and SPSS13.0 software (IBM, Armonk, NY, USA). The significance of differences among different experimental groups was evaluated based on a one–way analysis of variance (ANOVA). Finally, significant differences were evaluated using an independent sample *T*-test (*p* = 0.05) or a Fischer’s *F*-test (*p* = 0.01 or 0.05). 

## 3. Results and Discussion

### 3.1. Influence of Different Extraction Parameters on the Extraction Rate of ORPs

As a traditional extraction method for polysaccharides from natural products, the hot extraction method offers many advantages, including low cost, convenient operation, and no special equipment required, over other separation methods. Thus, it is suitably used for large–scale industrial applications [28].

It has been confirmed that natural polysaccharides could be extracted with alkaline solutions to maximize material utilization and expand the extraction yield. To determine the effects of various alkali concentrations on the extraction rates of ORPs, the extraction was conducted at alkali concentrations of 0.01, 0.02, 0.03, 0.04, 0.05, and 0.06 mol/L, and other conditions were kept as below: a ratio of material to liquid of 1: 100.0 g/mL, an extraction temperature of 70.0 °C, and an extraction time of 4.0 h. The results showed that the extraction rate of ORPs dramatically increased (*p* < 0.05) with the increase in alkali concentration (from 0.01 to 0.02 mol/L), and reached its maximum (15.8% ± 0.9%) when the alkali concentration was 0.02 mol/L. However, as the alkali concentrations further increased (from 0.03 to 0.06 mol/L), the extraction rate significantly decreased (*p* < 0.05) and then gradually became constant, as shown in Figure 1a. It has been confirmed that low concentrations of alkaline solution could contribute to destructing the cell wall, which is beneficial for the extraction of natural polysaccharides. However, excessive alkali may lead to an esterification reaction of the polysaccharides, which would cause a decrease in the extraction rate [29,30]. The results were consistent with the yield of okra and *Tremella fuciformis* polysaccharides prepared by alkali water extraction (10.1~20.2%) [31,32].

The influence of the ratios of different materials to liquid on the extraction rate of ORPs is shown in Figure 1b, with other extraction parameters being an alkali concentration of 0.01 mol/L, an extraction temperature of 70.0 °C, and an extraction time of 4.0 h. The ratio of material to liquid seemed to significantly affect the extraction rate of ORPs. The results indicate that the extraction rate rose with of the increase in the ratio of material to liquid. As the ratio rose to 1: 120.0 g/mL, the extraction rate reached 15.7 ± 1.3%). When we further increased the ratio of material to liquid, no remarkable changes were observed in the extraction rate of the ORPs.

To investigate the influence of temperature on the extraction rate of ORPs, the extraction temperature was set to 50.0, 60.0, 70.0, 80.0, and 90.0 °C, respectively. All other conditions were set as follows: alkali concentration of 0.01 mol/L, ratio of material to liquid of 1: 100.0 g/mL, and extraction time of 4.0 h. Figure 1c shows the effect of different temperatures on the extraction rate. It can be seen that temperature is also an important factor in the extraction of ORPs. The extraction rate was enhanced from 50.0 to 70.0 °C, and then gradually decreased with the further rise in extraction temperature. When the temperature reached 70.0 °C, the extraction rate was raised to its maximum value (14.5 ± 1.7%). 

Finally, the effects of various extraction times on the extraction rate of ORPs is indicated in Figure 1d. On the basis of the preparatory experiment, the extraction time was kept at 2.0, 3.0, 4.0, 5.0, and 6.0 h, respectively. All of the other conditions were adjusted as follows: alkali concentration of 0.01 mol/L, ratio of material to liquid of 1: 100.0 g/mL, and extraction temperature of 70.0 °C. The extraction rate was enhanced with the increase in extraction time. When the extraction time was kept as 4 h, the extraction rate reached 15.4 ± 1.1%. With a longer extraction time, no noticeable change was found in the extraction rate of ORPs.

To summarize, the alkali extraction method can be applied to enhance the extraction efficiency and reduce the separation time of acidic ORPs, but the amount of alkali added should be paid attention to and strictly controlled. In addition, common influencing factors should not be overlooked either, including the material–to–liquid ratio, temperature, and time.

### 3.2. Optimization of Extraction Parameters of ORPs 

#### 3.2.1. Model Fitting and Adequacy Checking

The test data were processed according to the multiple regression analysis, as given below. A second–order polynomial modeling method worked out the regression analysis on the basis of the relationship between the dependent response variable (*X*) and the independent response variable (*Y*): *Y* = 15.74 + 0.18*X*_1_ + 1.02*X*_2_ − 0.34*X*_3_ − 0.21*X*_4_ + 1.45*X*_1_*X*_2_ − 1.06*X*_1_*X*_3_ − 0.45*X*_1_*X*_4_ − 1.92*X*_2_*X*_3_ + 1.09*X*_2_*X*_4_ + 1.06*X*_3_*X*_4_ − 2.32*X*_1_^2^ − 1.72*X*_2_^2^ − 2.53*X*_3_^2^ − 0.73*X*_4_^2^
where *X*_1_ is the alkali concentration (mol/L), *X*_2_ is the ratio of material to liquid (g/mL), *X*_3_ is the extraction temperature (°C), and *X*_4_ is the extraction time (h). 

Based on the ANOVA results for this model indicated in Table 2, the *F* value was 270.65 and *p* < 0.0001, which suggested that the model’s response to the extraction rate of ORPs was highly significant. In addition, the *p* value of lack of fit in this model was 0.6515 (*p* > 0.05), which suggested that the model fit well. Thus, the reliability of the model was verified. Furthermore, the values of coefficient of determination (*R*^2^) and the adjusted determination coefficient (*R*^2^_adj_) indicated that the model fit well with the experimental data and the theoretical values of the extraction rate of ORPs, which were 0.9963 and 0.9926, respectively. Furthermore, the lack of fit (*p* > 0.05) suggested that there a failure occurred with a model representing the test data. The chances of a “lack of fit–value” due to noise were remote. As a result, the results indicated a good fit. An extremely low coefficient of variation value (*C.V.* = 16.16%) represented the high reproducibility and reliability of the experimental values. As shown in Table 2, the linear coefficients (*X*_1_, *X*_2_
*X*_3_, and *X*_4_); quadratic term coefficients (*X*_1_^2^, *X*_2_^2^, *X*_3_^2^, and *X*_4_^2^); and cross–product coefficients (*X*_1_*X*_2_, *X*_1_*X*_3_, *X*_1_*X*_4_, *X*_2_*X*_3_, *X*_2_*X*_4_, and *X*_3_*X*_4_) were all significant (*p* < 0.05). The *F* value showed that the order of factors influencing the ORP extraction rate was ratio of material to liquid > extraction temperature > extraction time > alkali concentration, and the order of the interaction effects was *X*_2_*X*_3_ > *X*_1_*X*_2_ > *X*_2_*X*_4_ > *X*_1_*X*_3_ >*X*_3_*X*_4_ > *X*_1_*X*_4_.

#### 3.2.2. Response Surface Analysis

In order to directly observe the influence of independent variables on the response, three–dimensional (3D) response surface plots were obtained for the purpose of quantifying the optimal values in order to achieve the optimal extraction rate of ORPs. Both the 3D response surface plots (Figure 2) and contour plots (Figure 3) indicated the interaction influence of the various variables, and there was a intuitionistic interpretation of the different influences in the steep degree of the 3D plot. The steepness of the response surface represented the degree of interaction effects of two experimental variables.

The influences of alkali concentration and the ratio of material to liquid; alkali concentration and extraction temperature; and alkali concentration and extraction time on the extraction rate are shown in Figure 2a–c and Figure 3a–c, respectively. The above results indicate that the influence of alkali concentration on the extraction rate of ORPs was greater than that of extraction temperature, extraction time, and the ratio of material to liquid on the curved surface. The influences of alkali concentration on the extraction rate were significant. The extraction rate of ORPs rose rapidly when the alkali concentration increased from 0.01 to 0.03 mol/L. However, with the alkali concentration increasing from 0.03 to 0.04 mol/L; the ratio of material to liquid from 1:112.0 to 1:120.0 g/mL; the extraction temperature from 75.0 to 80.0 °C; and the extraction time from 4.5 to 5.0 h, the extraction rate significantly decreased.

The effects of the ratio of material to liquid and extraction temperature and the ratio of material to liquid and extraction time on the extraction rate are illustrated in Figure 2d,e and Figure 3d,e. These results indicate that the influence of the ratio of material to liquid on the extraction rate rose more rapidly than that of the extraction temperature and time, since the curved surface of the ratio of material to liquid was much sharper than the curved surface of the latter. The extraction rate went up quickly with the enhancement of the ratio of material to liquid from 1:80.0 to 1:120.0 g/mL, but rose slowly with the increase in extraction temperature from 60.0 to 75.0 °C and in extraction time from 3.0 to 4.0 h. This could be explained by the fact that the larger ratio of material to liquid facilitated a higher leaching–out rate of ORPs into the alkali solution, which caused an increase in the extraction yield.

As indicated in Figure 2f and Figure 3f, the effects of extraction temperature and time on the extraction rate were investigated. The extraction rate went up slightly with the increase in the levels of these interaction factors. The significance of the interaction between the variables was examined based on the steepness of the response surface, and results similar to those of the ANOVA test were found.

#### 3.2.3. Model Verification

The optimal conditions for maximizing the extraction rate of ORPs were obtained from the model: alkali concentration, 0.02 mol/L; ratio of material to liquid, 1:112.7 g/mL; extraction temperature, 66.0 °C; and extraction time, 4.0 h. The predicted point was experimentally verified under the actual optimum conditions: alkali concentration of 0.02 mol/L, ratio of material to liquid of 1:113.0 g/mL, temperature of 66.0 °C, and time of 4.0 h. Under the optimum conditions, the extraction rate of ORPs was 16.2 ± 0.1%, and no significant differences were found with which to compare the predicted value of 16.3% (*p* > 0.05). All of the results indicate that the experimental model used in this study was effective.

### 3.3. Preparation and Physiochemical Characteristics

An extract, in an alkaline solution of polysaccharides from *O. raphanipies,* was obtained based on the optimal extraction parameters. It was concentrated, precipitated with ethanol (95%, *v*/*v*), dialyzed, deproteinated, and freeze–dried to prepare the crude ORPs, and ORPs–I~ORPs–V was obtained by precipitation with different concentrations of ethanol (20%, 40%, 60%, 80%, and 80% above, *v*/*v*), respectively. Because the crude polysaccharides may contain various impurities, including protein, monosaccharide, and other small molecule compounds, ORPs–V was further purified by the ion–exchange column DEAE–52 and Sephadex G–150 columns. As shown in Figure 4a,b, there was one dominant peak in the elution, which was obtained by elution with 0.15 mol/L NaCl solution on the DEAE–52 cellulose column. The major fraction was further purified by Sephadex G–150 column, indicating that ORPs–V was a homogeneous polysaccharide [33,34].

As an important physicochemical property, *Mw* significantly affects the biological activities of polysaccharides. HPGPC was used to determine the *Mw* of ORPs–V. According to the equation of standard curve, Log *Mw* = −3.5369 *T* + 33.437 (*R*^2^ = 0.9994), (*Mw* was the weight–average molecular weight, and *T* represented the retention time). The *Mw* of ORPs–V was calculated to be 18.86 kDa, and the retention time was 20.35 min. As shown in the HPGPC spectrum, a single and symmetrical peak for ORPs–V was obtained, indicating that the purified ORPs–V was homogeneous polysaccharide (Figure 4c) [35]. 

### 3.4. FT–IR Spectrometry

As a practical qualitative analysis tool of functional groups of natural active ingredients, infrared spectroscopy is applied to clarify polysaccharide structure. FT–IR of ORPs–I~ORPs–V was performed, and the results were shown in Figure 5a. The ORPs had the characteristic peaks of polysaccharides, including peaks at 3460, 2937, 1747, 1666, 1404, 1309, 1049, and 790 cm^−1^. However, the ORPs–V had a broadly-stretched intense peak at 3460 cm^−1^, which was usually considered to be the O–H stretching vibration. A relatively weak peak at 2937 cm^−1^, which can be ascribed to the asymmetric bending vibration caused of C–H bond. The absorption at 1666 cm^−1^ may be attributed to the asymmetric stretching vibration of the C=O bond, while the peak at 1404 cm^−1^ could be considered as the symmetric stretching vibration of the C–H bond. In addition, a weak peak that appeared at 1747 cm^−1^ was attributed to the C=O valent vibration of the O–acetyl group. These results confirmed the presence of uronic acid in ORPs, which had been confirmed by the results of the monosaccharide composition analysis. The notable absorption between 1000 and 2000 cm^−1^ was caused by the overlapping of ring vibration, the C–O–C glycosidic band vibrations, and the stretching vibrations of the C–O–H side groups, indicating the possible presence of pyranose. The signal at around 790 cm^−1^ was attributed to the absorption of *α*–type glycosidic linkage, and the signals in the range of 800 to 900 cm^−1^ were attributed to the presence of *β*–configuration. The above results demonstrated nearly no difference between the FT–IR signals of ORPs [36,37,38]. 

### 3.5. Monosaccharide Composition

Previous studies have suggested that the biological activities of polysaccharides are closely related to their monosaccharide compositions. Natural polysaccharides with a larger quantity of uronic acids are usually considered to possess superior biological activities. The sugar residues of ORPs were measured by comparison of the relative retention times with those of standard monosaccharides (Figure 4d), which is indicated in Table 3. The results suggest that the ORPs contained L–fucose, L–rhamnose, L–arabinose, D–glucose, D–mannose, D–galactose, D–xylose, D–fructose, D–galacturonic acid, and D–glucuronic acid. The L–fucose, D–glucose, D–galactose, D–xylose, D–fructose, and D–glucuronic acid were the principal components of ORPs–I and ORPs–II; while the L–arabinose, D–glucose, D–galactose, and D–fructose were the principal components of ORPs–III. In addition, the molar ratio of L–fucose, L–rhamnose, L–arabinose, D–glucose, D–galactose, D–mannose, D–xylose, and D–fructose in ORPs–V was 1.73:1.20:1.13:2.87:8.71:2.89:1.42:0.81. The (Gal + Ara)/Rha ratio in ORPs–V was 8.20, and the higher ratio of (Gal + Ara)/Rha indicated that there were many side chains composed of neutral sugar in ORPs–V. Most notably, the percentages of glucose, galactose, and mannose were higher than those of other monosaccharides, which may be positively correlated to the antioxidant activities of ORPs. The above results are similar to those of the study on *Camellia fascicularis* and *Lycium barbarum* by Peng et al. and Zhang et al. [39,40]. In addition, glucuronic acid, galactose, and arabinose content may also play an equally important role in the antioxidant activities of polysaccharides [41]. However, the anti–oxidant activities of natural polysaccharides were also highly influenced by the branching degree, order of sugar units, and glycosidic bonds, which cause the influence of monosaccharide composition to be insufficient for them [42].

### 3.6. NMR Analysis

According to the results of the FT–IR analysis, there were no apparent differences in the ORPs. Thus, only the ORPs–V fraction was further investigated using ^1^H–NMR, ^13^C–NMR, and HSQC spectroscopy to further elucidate the structural properties of ORPs. The ^1^H–NMR, ^13^C–NMR, and HSQC spectra of ORPs–V are shown in Figure 5b–d. As shown in Figure 5b,c, the ^1^H–NMR and ^13^C–NMR spectra signals of ORPs-V were mainly concentrated in the regions of δ 3.00~5.50 ppm and 60.00~110.00 ppm, which is typical of polysaccharides [34,43,44]. The ^1^H–NMR spectrum was frequently used to identify the glycosidic bond conformation in the polysaccharide; the anomeric protons with δ 5.00~5.80 ppm were in *α*–configuration, and the anomeric protons with δ 4.40~5.00 ppm were in *β*–configuration [45]. In the ^1^H–NMR spectrum, there was a strong signal at 4.70 ppm, which could be assigned to D_2_O. The heterotopic hydrogen proton signal appeared at 4.90~5.20 ppm, indicating that the ORPs–V was pyranose. Three small high–field signals at 4.43, 4.47, and 4.50 ppm were considered to suggest that there was a *β*–anomeric configuration in the ORPs–V, which was consistent with the results of the FT–IR analysis [46]. The signals in the region of 2.00~2.40 ppm suggested that some acetyl groups were binding in the ORPs–V, which agreed with both the literature and the FT–IR results (signal peak at 1730 cm^−1^). The ^13^C–NMR spectrum of ORPs–V had a clearer chemical shift and better resolution than ^1^H–NMR. In addition, the wide signal at 3.50~4.00 ppm was attributed to galactose residues, and the signal peaks between 3.35 and 4.40 ppm were regarded as the protons of the CH_2_–O and CH–O groups on the sugar rings. The signal peaks at 4.53, 4.93, and 5.02 ppm were due to mannitol and glucose [47]. 

The δ 60.00~85.00 ppm value was attributed to the chemical signal peaks of C_2_ to C_6_, and δ 90.00~110.00 ppm was the signal for the anomeric carbon area. ORPs–V contained three major types of anomeric carbon signals. The anomeric carbon signals at δ 100.08, 102.16, and 103.48 ppm were observed in the ^13^C–NMR spectrum. The lower–field chemical signals, in the range of 175.89 ppm, showed that there were some uronic acids in the ORPs. The overlapped signals of C_2_~C_6_ were seen in the range of δ 60.00~80.00 ppm. No absorbance at δ 80.00~90.00 ppm was observed in the ^13^C–NMR spectrum, which indicated that there was no furanose in ORPs–V. The chemical shifts of anomeric protons below 4.77 ppm and carbon resonance above 100.45 ppm indicated that the Gal*p* residues were *β*–linked–anomeric [48].

Identifying the chemical linkages in the natural polysaccharides was highly difficult, because the signal peaks of the fractions overlapped severely in the ^1^H–NMR and ^13^C–NMR spectra [49]. As a result, HSQC ^13^C/^1^H–NMR was applied to further evaluate the structural properties of the ORPs–V. As indicated in Figure 5d and Table 4, five predominant signals were observed in HSQC ^13^C/^1^H cross peaks at 102.71/4.37, 71.07/3.43, 78.23/3.64, 71.54/3.51, and 69.19/3.93 ppm, which were seen as the C_1_–H_1_, C_2_–H_2_, C_3_–H_3_, C_4_–H_4_, and C_5_–H_5_ of the *β*–D–galactose residues, respectively. Some cross–peaks could be attributed to Xyl*p* units, which were also observed in the HSQC spectrum. For instance, the spectra indicated the presence of HSQC ^13^C/^1^H cross peaks at δ 73.70/3.42 and 61.60/3.78 ppm, which may be attributed to C_3_ and C_5_ of the *β*–1, 3, and 4–linked Xyl*p* residues. The structural elements of the *β*–T–linked Xyl*p* units were observed in the spectra of HSQC at δ 99.10/4.48 (C_1_–H_1_), 72.00/3.31 (C_2_–H_2_), and 69.20/3.62 (C_4_–H_4_) [50]. Nevertheless, due to the serious overlap of the NMR signals and the relatively low proportions of xylose and 2, 6–linked mannose units, it is challenging to assign them NMR signals [51]. Further investigations are needed to elucidate, in depth, its detailed structural features, including methylation analysis, NOESY, COSY, TOCSY, HMQC, and HMBC. 

### 3.7. Anti–Oxidant Activities In Vitro

Free radicals pose a major risk to the health of the human body and play a vital role in different kinds of lifestyle–related diseases, including aging, heart disease, cancer, arteriosclerosis, and diabetes. Antio–xidants are crucial because they can eliminate free radicals from the body, which may harm DNA and proteins. Polysaccharides are generally regarded as the main active ingredients in mushrooms. The anti–oxidant activities of the ORPs–I~ORPs–V were evaluated in vitro by ABTS^·+^, hydroxyl, DPPH radial, and reducing power tests, which are displayed in Figure 6. 

ABTS^+^–scavenging assays have been generally applied to test the anti–oxidant capacity of natural polysaccharides. ABTS^+^ could react with K_2_(SO_4_)_2_ to form stable glaucous cation radicals, which would be measured at 734 nm [52]. As indicated in Figure 6a, the scavenging activities of the ORPs–I~ORPs–V against ABTS^·+^ were enhanced in a dose–response pattern, which indicated the presence of significant scavenging activity at all concentrations. Compared to other ORPs, ORPs-V showed better scavenging activity, second only to Vc. The EC_50_ values of ORPs–I, ORPs–IV, and ORPs–V were far lower than 0.5 mg/mL, respectively. However, both ORPs-II and ORPs-III showed relatively weaker scavenging activity, and their EC_50_ was approximately 0.5 mg/mL. The hydroxyl radical–scavenging activities of the ORPs are shown in Figure 6b, and the EC_50_ values of ORPs–IV and ORPs–V were significantly lower than 0.5 mg/mL. Both ORPs–II and ORPs–III showed weaker hydroxyl radical–scavenging activities, because the EC_50_ values of both were greater than 1.0 mg/mL. It was noteworthy that ORPs–I~ORPs–V showed significant scavenging activities for DPPH, but their EC_50_ value was much lower than 0.5 mg/mL (Figure 6c). The reducing power is another key index with which to evaluate the antioxidant activities of natural products. The conversion of iron (III) to iron (II) with ORPs is shown in Figure 6d. All samples possessed a certain reducing power with a dose–response pattern. Notably, the ORPs–V presented significantly stronger reducing power than the other samples. At 1.0 mg/mL, the ORPs–V had significant reducing power, greater than 1.5 in absorbance. 

Taken together, the EC_50_ of ORPs–V on ABTS^+^, hydroxyl, DPPH radial, and reducing power was the lowest; as a result, the antioxidant abilities of ORPs–V in vitro were stronger than other fractions. The free radical scavenging activities of polysaccharides from mushrooms are primarily attributed to the existence of reducing hemiacetal hydroxyl, which can absorb electrons to reduce free radical chain reactions and directly undergo redox reactions with peroxides and active oxygen, thus preventing the generation of free radicals. In addition, the total reducing power of ORPs is closely related to the power of providing electrons.

### 3.8. Anti–Oxidant Activities In Vivo

Oxidative stress has been widely considered as a central player in various chronic metabolic diseases, including obesity, diabetes, cancer, cardiovascular disorders, and inflammatory diseases [53]. SOD, CAT, GSH–Px, and LDH are the main indicators for evaluating the anti–oxidant capacity of experimental animals, as both their activities and the balance between them play important roles in reducing the oxidative damage [54,55]. To validate the results regarding the anti–oxidation activity of ORPs–V in vitro, both enzymatic activities (SOD, CAT, GSH-Px, and LDH) and MDA content in the serum of the HFD model mice were determined. As shown in Figure 7, remarkable decreases in SOD, CAT, GSH–Px, and LDH activities, but significant increases in MDA content, were observed in mice of the HFD model control group when compared to those in the NC group (*p* < 0.05), indicating that the HFD model mice had been under severe oxidative stress caused by their high–fat diets. Interestingly, after the treatment with ORPs–V at a dose of 200 mg/kg/d, the enzyme activities of SOD, CAT, GSH–Px, and LDH in the ORPs–V–H group were significantly increased to 29.5 ± 0.4, 7.9 ± 0.3, 11.1 ± 0.3, and 68.7 ± 0.9 U/mL, respectively, when compared with those in the HFD model control groups (*p* < 0.01). Most notably, the enzyme activities of GSH–Px in the ORPs–V–H group were significantly increased, markedly higher than the NC group (*p* < 0.01). In addition, the contents of MDA in the ORPs–L, ORPs–M, and ORPs–V groups were markedly reduced when compared with those in the HFD model control groups (*p* < 0.01). Notably, no significant differences were found between the ORPs–V–H group and the NC group (*p* > 0.05). The above results indicate that ORPs–V can effectively alleviate HFD–induced oxidative stress, and a high dose of ORPs–V performs better activities than a low dose. More in–depth studies should be conducted to elucidate the detailed mechanism behind this effect.

To summarize, these results indicate that the ORPs–I~ORPs–V have different degrees of anti–oxidant activities in a concentration–dependent manner, and the experimental results of four assays in vitro were similar. The anti–oxidant activities of ORPs–V with lower *Mw* were better than those of other fractions, and the anti–oxidant activities of ORPs–V in vivo were also confirmed. Interestingly, ORPs–V with higher contents of glucose, galactose, and mannose seemed to have better anti–oxidant activities. Previous studies have shown that *Mw* in polysaccharides may be an important factor in anti–oxidant activities. Moreover, monosaccharide composition, type of glycosidic linkage, the amount of hydroxyl group, or chain conformation of polysaccharides seem to be closely correlated to polysaccharides’ anti–oxidant activities. For instance, the polysaccharides with galactose, glucose, and glucuronic acid showed stronger anti–oxidant activities than those without them [50]. The polysaccharides with higher arabinose, rhamnose, and galactose concentrations showed superior anti–oxidant activities [56]. In addition, the higher content of neutral monosaccharides seemed to negatively affect the anti–oxidant activities of the polysaccharides, but high acidic monosaccharide contents may have enhanced their anti–oxidant activities [57]. The monosaccharides in ORPs might play a significant role in the anti–oxidant activities of ORPs. Additionally, it was also suggested that polysaccharides’ anti–oxidant properties may be related to hydroxymethylfurfural content. Further research on the relationships among the degree of *Mw*, the branch, the microstructure, and the anti–oxidant activities of ORPs remains to be carried out.

## 4. Conclusions

As a precious edible mushroom with a high nutritional value, which is mainly distributed in Hainan, Fujian, Jiangsu, and Yunan in China, ORPs are attracting great interest from researchers. In this study, the extraction rate of the ORPs was highly improved to 16.2% under the optimal conditions (alkali concentration of 0.02 mol/L, ratio of material to liquid of 1: 112.7 g/mL, extraction temperature of 66.0 °C, and extraction time of 4.0 h). All responses were statistically studied and found to fit with the quadratic regression model. The ratio of material to liquid was found to have the main effect on the extraction rate of ORPs, which was more important than the extraction temperature, extraction time, and alkali concentration. The anti–oxidant activities of ORPs, both in vitro and in vivo, were evaluated and classified using water extraction and ethanol precipitation methods. ORPs–V displayed the strongest antioxidant activity in vitro, and the animal experiments also confirmed the anti–oxidant activities of ORPs–V. In addition, the FT–IR, monosaccharide analysis, and NMR showed that ORPs–V consisted of L–fucose, L–rhamnose, L–arabinose, D–glucose, D–galactose, D–mannose, D–xylose, D–fructose, D–galacturonic acid, and D–glucuronic acid. The main backbone of the proposed AAP–V structure was composed of 1→3, 6 and 1→4, 6–*β*–Gal, with branches consisting of *β*–Glc*p*–(1→6), *β*–Man*p*–(1→6), *β*–Man*p*–(1→2), Ara*f*–(1→3, 5), and *β*–Xyl*p*–(1→). However, their molar ratio was significantly different, and the high content of glucose, galactose, and mannose seems to have had a positive effect on their anti–oxidant activities. Further studies should be undertaken to elucidate the structure–function relationship and molecular mechanism of ORPs and ORPs–V. These results indicate that ORPs–V may be used as a source of natural anti–oxidants in the food or pharmaceutical industries, and can provide an important reference for the comprehensive utilization of *O. raphanipies*. 

## Figures and Tables

**Figure 1 polymers-15-02917-f001:**
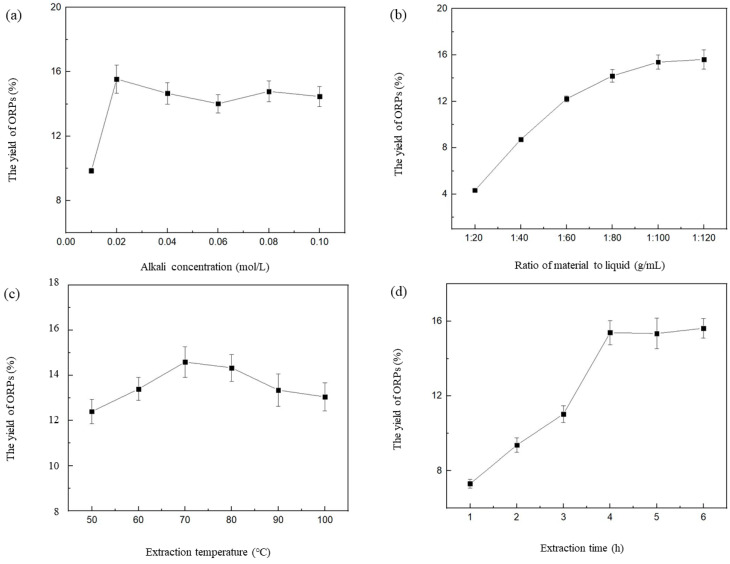
Effect of different extraction parameters on the extraction rate of ORPs. (**a**) Alkali concentration, mol/L; (**b**) ratio of material to liquid, g/mL; (**c**) extraction temperature, °C; (**d**) extraction time (h). All of these values represent the mean ± SD of triplicate results.

**Figure 2 polymers-15-02917-f002:**
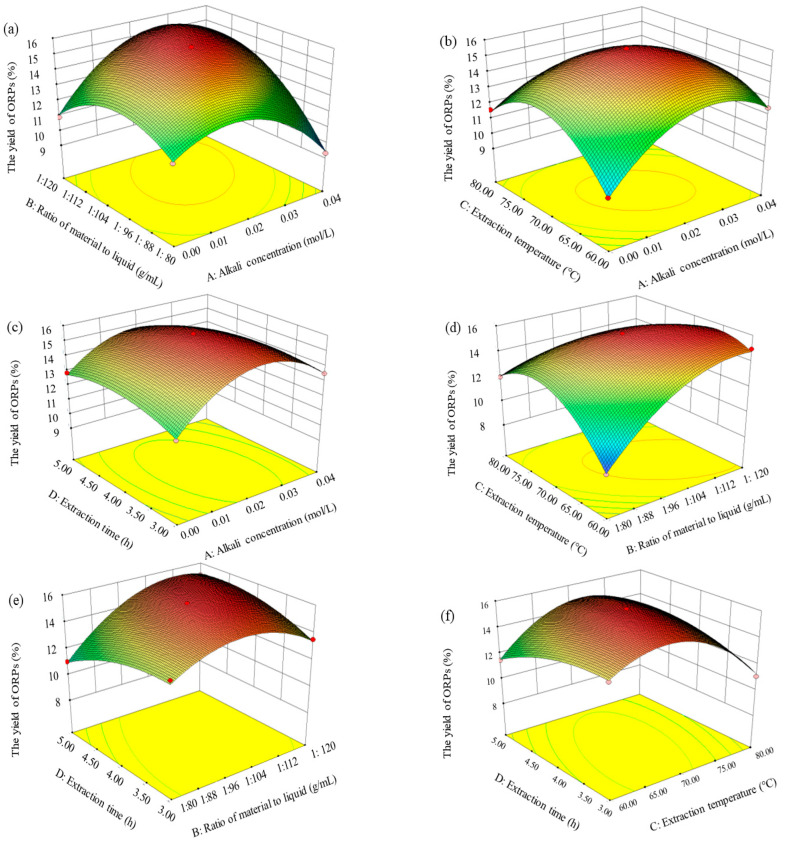
Response surface plots showing the interactions between different selected factors. (**a**) Effect of alkali concentration and ratio of material to liquid on extraction rate of ORPs; (**b**) effect of alkali concentration and extraction temperature on extraction rate of ORPs; (**c**) effect of alkali concentration and extraction time on extraction rate of ORPs; (**d**) effect of ratio of material to liquid and extraction temperature on extraction rate of ORPs; (**e**) effect of ratio of material to liquid and extraction time on extraction rate of ORPs; and (**f**) effect of extraction temperature and extraction time on extraction rate of ORPs.

**Figure 3 polymers-15-02917-f003:**
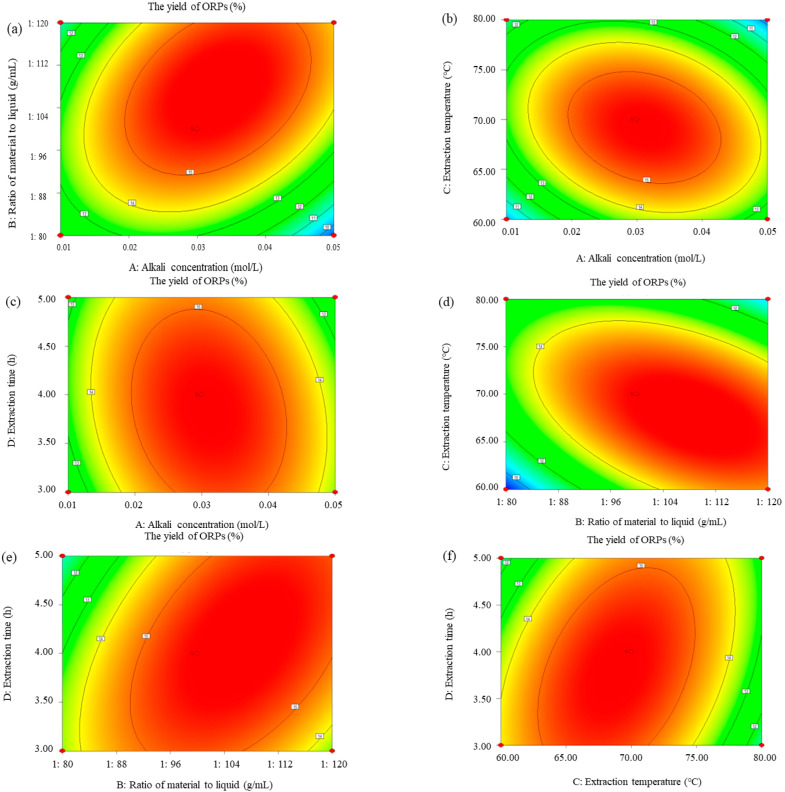
Contour plots displaying the effects of different extraction parameters on the extraction rate of ORPs. (**a**) Effect of alkali concentration and ratio of material to liquid on extraction rate of ORPs; (**b**) effect of alkali concentration and extraction temperature on extraction rate of ORPs; (**c**) effect of alkali concentration and extraction time on extraction rate of ORPs; (**d**) effect of ratio of material to liquid and extraction temperature on extraction rate of ORPs; (**e**) effect of ratio of material to liquid and extraction time on extraction rate of ORPs; and (**f**) effect of extraction temperature and extraction time on extraction rate of ORPs.

**Figure 4 polymers-15-02917-f004:**
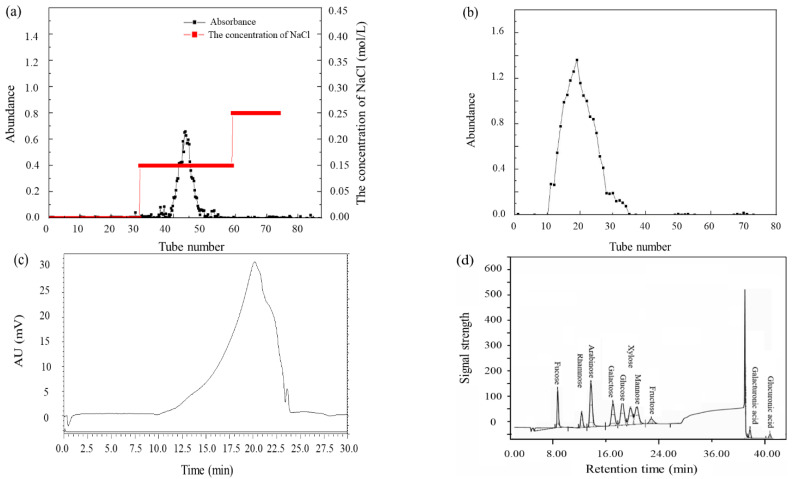
The purification of ORPs–V. (**a**) The elution curves of ORPs–V with a DEAE–52 cellulose column; (**b**) The elution curves of ORPs–V with Sephadex G–150; (**c**) The molecular weight distribution of ORPs–V and (**d**) GC–MS chromatogram of 10 standard monosaccharides.

**Figure 5 polymers-15-02917-f005:**
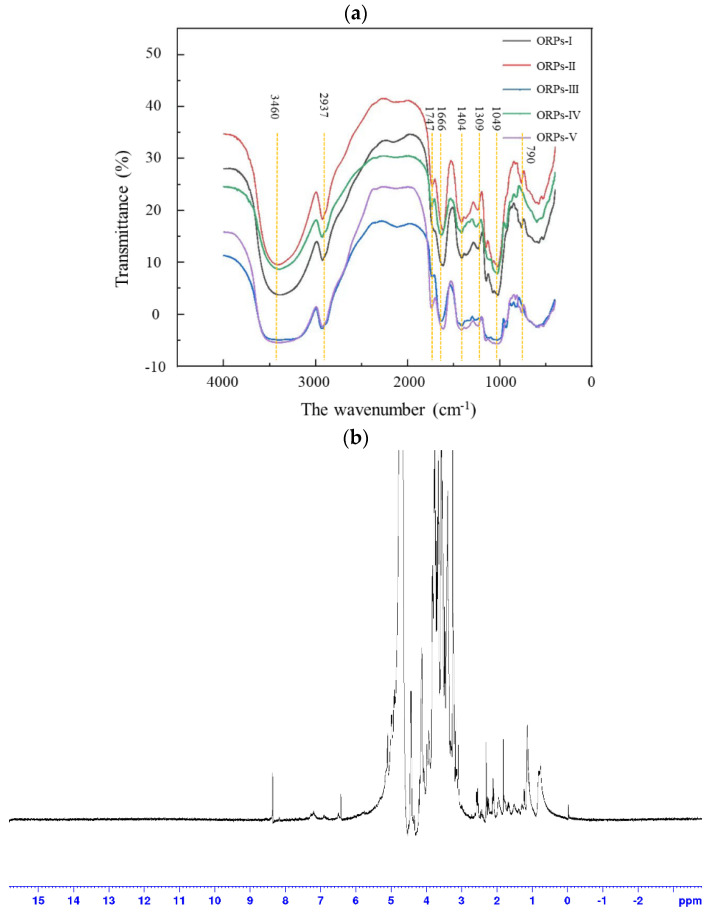
(**a**) FT–IR spectra of ORPs; (**b**) ^1^H–NMR spectrum of ORPs–V; (**c**) ^13^C–NMR spectrum of ORPs–V; (**d**) ^1^H–^13^C HSQC spectrum of ORPs–V.

**Figure 6 polymers-15-02917-f006:**
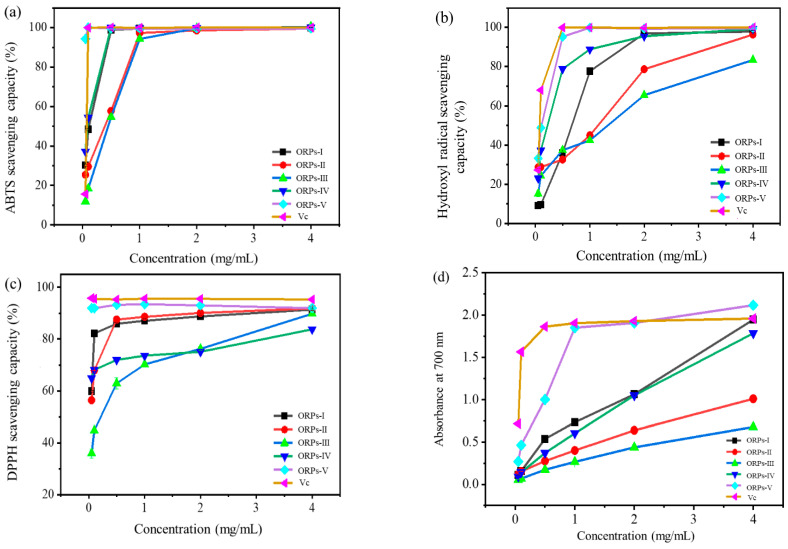
In vitro anti–oxidant activities of ORPs–I~ORPs–V. (**a**) ABTS radical cation scavenging activity; (**b**) hydroxyl radical scavenging activity; (**c**) DPPH radical scavenging activity; and (**d**) reducing power.

**Figure 7 polymers-15-02917-f007:**
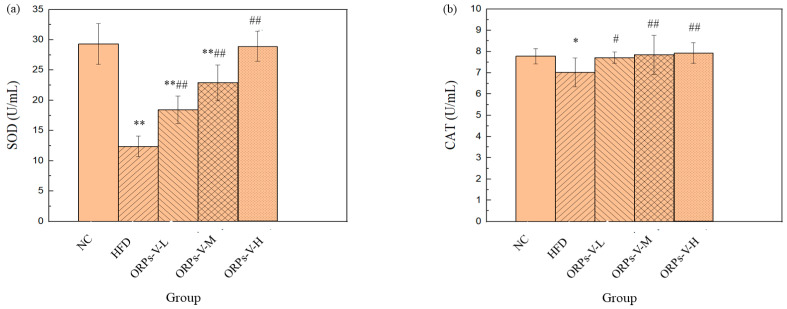
Effects of ORPs–V on SOD activity (**a**); CAT activity (**b**); GSH–Px activity (**c**); LDH activity (**d**); and MDA content (**e**) in serum. The values are the mean ± SD of six mice per group. ** *p* < 0.01, * *p* < 0.05, compared with the NC group; ## *p* < 0.01, # *p* < 0.05, compared with the HFD model control group.

**Table 1 polymers-15-02917-t001:** Four–factor central composite design matrix and the response values for the extraction rate (%).

Run	Alkali Concentration (mol/L) *X*_1_	Ratio of Material to Liquid (g/mL) *X*_2_	Extraction Temperature (°C) *X*_3_	Extraction Time (h) *X*_4_	ORPs Yield (%)
1	0.02	1:120	80	4	10.4
2	0.02	1:100	70	4	15.7
3	0.02	1:80	70	3	13.8
4	0.02	1:120	70	5	15.1
5	0.02	1:100	60	3	14.0
6	0.03	1:100	70	5	12.4
7	0.02	1:120	60	4	15.0
8	0.03	1:120	70	4	14.2
9	0.03	1:100	80	4	9.8
10	0.02	1:120	70	3	13.5
11	0.02	1:100	70	4	15.7
12	0.02	100	70	4	15.7
13	0.02	80	60	4	8.8
14	0.01	100	80	4	11.6
15	0.02	80	80	4	12.0
16	0.01	100	70	3	12.2
17	0.03	100	60	4	12.5
18	0.02	80	70	5	11.0
19	0.01	100	70	5	12.9
20	0.03	80	70	4	9.3
21	0.01	80	70	4	11.9
22	0.02	100	80	5	12.8
23	0.01	100	60	4	10.0
24	0.03	100	70	3	13.5
25	0.01	120	70	4	10.9
26	0.02	100	60	5	11.5
27	0.02	100	80	3	11.1
28	0.02	100	70	4	15.7
29	0.02	100	70	4	15.7

**Table 2 polymers-15-02917-t002:** Analysis of variance (ANOVA) results for the regression parameters.

Parameter	Sum of Square	df	Mean Square	*F*-Value	*p*-Value	Significance
Model	123.25	14	8.80	270.65	<0.0001	**
*X* _1_	0.400	1	0.40	12.4	0.0034	**
*X* _2_	12.36	1	12.36	380.08	<0.0001	**
*X* _3_	1.39	1	1.39	42.65	<0.0001	**
*X* _4_	0.52	1	8.097	16.01	0.0013	**
*X* _1_ *X* _2_	8.44	1	8.44	259.45	<0.0001	**
*X* _1_ *X* _3_	12.1452	1	4.49	138.18	<0.0001	**
*X* _1_ *X* _4_	0.3306	1	0.82	25.18	0.0002	**
*X* _2_ *X* _3_	64.2402	1	14.78	454.53	<0.0001	**
*X* _2_ *X* _4_	35.1056	1	4.71	144.77	<0.0001	**
*X* _3_ *X* _4_	15.4056	1	4.47	137.53	<0.0001	**
*X* _1_ ^2^	34.89	1	34.89	1072.61	<0.0001	**
*X* _2_ ^2^	19.23	1	19.23	591.12	<0.0001	**
*X* _3_ ^2^	41.66	1	41.66	1280.70	<0.0001	**
*X* _4_ ^2^	3.47	1	3.47	106.76	<0.0001	**
Residual	0.46	14	0.033			
Lack of fit	0.46	10	0.46	0.4931	0.6515	-
Pure error	0.00	4	0.00			CV% = 16.16
R-Squared	0.9963		Adi.R-squared	0.9926		

** significant at *p* < 0.01. Source: ANOVA using Design-Expert 7.0.0.

**Table 3 polymers-15-02917-t003:** The results of our monosaccharide composition analysis of ORPs.

Samples	Fuc	Rha	Ara	Glc	Gal	Man	Xyl	Fru	GalA	GlcA
ORPs–I	3.55	0.69	1.43	4.70	6.81	0.14	5.27	2.78	–	3.91
ORPs–II	4.00	0.64	1.61	7.85	8.63	2.45	3.17	1.96	–	1.79
ORPs–III	0.02	–	0.19	0.29	0.35	–	–	1.07	–	–
ORPs–IV	–	–	0.01	0.01	0.03	0.01	0.02	0.10	0.01	14.73
ORPs–V	1.73	1.20	1.13	2.87	8.71	2.89	1.42	0.81	0.60	0.90

–: Not detected. Fuc, fucose; Rha, rhamnose; Ara, arabinose; Glc, glucose; Gal, galactose; Man, mannose; Xyl, xylose; Fru, fructose; GalA, galacturonic acid; GlcA, glucuronic acid.

**Table 4 polymers-15-02917-t004:** The chemical shift assignments of ^1^H–NMR and ^13^C–NMR spectra of ORPs–V.

Residues	Sugar Linkage	H1/C1	H2/C2	H3/C3	H4/C4	H5/C5	H6/C6
*Gal*	*→3, 6)*–*β*–*Galp*–*(1*→	4.37 102.71	3.43 71.07	3.64 78.23	3.51 71.54	3.24 71.26	3.93 69.19
	*→4, 6)*–*β*–*Galp*–*(1→*	4.41 102.71	3.24 71.26	3.51 71.44	3.94 73.24	3.59 70.97	4.03 69.67
*Ara*	*T*–*Araf*–*(1*→	5.16 100.45	4.17 79.34	3.72 73.45	3.78 73.39	3.63 71.14	3.53 67.67
	*1, 3, 5*–*Araf*–*(1*→	5.23 100.45	4.25 79.34	3.77 73.39	3.63 71.14	3.51 67.67	3.41 n.d.
*Man*	→*6)*–*β*–*Manp*–*(1*→	4.41 103.20	3.55 70.80	3.82 71.50	3.98 68.30	3.96 72.60	4.09 68.00
	*1, 2*–*β*–*Manp*–*(1*→	5.14 102.00	4.16 79.00	3.97 71.30	3.73 69.40	3.94 74.50	3.93 64.50
*Xyl*	*T*–*β*–*Xylp*–*(1*→	4.45 102.16	3.30 73.09	3.48 75.52	3.51 73.46	n.d. n.d.	n.d. n.d.
		4.74 104.10	3.33 73.22	3.25 76.23	3.63 69.21	3.74 67.30	n.d. n.d.
*Glc*	*1, 6*–*β*–*Glcp*–*(1*→	4.48 102.81	3.29 73.10	3.45 75.64	n.d. n.d.	n.d. n.d.	n.d. n.d.

Note: n.d.: not detected.

## Data Availability

Some or all data during the study are available from the corresponding author by request.

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
