# Peer review of "Extraction, Isolation, Screening, and Preliminary Characterization of Polysaccharides with Anti–Oxidant Activities from *Oudemansiella raphanipies"

_polymers, 2023, doi:10.3390/polym15132917_

Round 1

Reviewer 1 Report

It is advisable to rethink the abstract in order to fully identify the contribution or novelty of the study. In this sense, the study is aimed at designing a good antioxidant agent, so this contribution must be supported and ratified in case this is the novelty of the study. In the same way, the title can be improved and pointed out thinking about the main contribution of the study, in the way in which it is proposed seems very general.

State of the art can be expanded.

The methodology is described broadly and adequately.

It is recommended to broaden the discussion regarding the effects of alkali concentrations on the extraction rate of ORPs. The analysis can be deepened with the appropriate scientific support, in order to clarify even more.

Correlate the different effects, with respect to the variables studied.

Section 3.3, if it is possible to improve the discussion, is not clear. And support with adequate scientific literature,

Improve quality of FTIR and other spectra.

Section 3.8, Improve analysis with adequate scientific support.

Conclusions can be improved, after expanding the analysis of results with adequate scientific support.

Author Response

To reviewer-1:

Response: Thank you for your efficient work in reviewing our manuscript. We are sorry for our carelessness.

(1) The title was revised as: “Extraction, isolation, screening, preliminary characterization of polysaccharides with antioxidant activities from Oudemansiella raphanipies”, and the abstract was revised, thank you for your recommendations and opinions.

(2) The discussion about the effects of alkali concentrations on the extraction rate of ORPs have been added, which is shown in Page 7 and 8 in yellow font. 

(3) The discussion has been added, and we support with a scientific literature, which is show in Page 12 and Page 22 in yellow font.

(4) The quality of FTIR, Figure 4, and Figure 6 has been improve, which is show Page 13, 14, and 18.

(5) The discussion has been added, and we support with a scientific literature, which is show in Page 19 and Page 23 in yellow font.

(6) The conclusions have been improved, which is show in Page 20 and Page 21 in yellow font.

Hope the revised version could be accepted. Thank you again for your suggestions.

Reviewer 2 Report

This manuscript presented a study about the extraction and characterization of Oudemansiella raphanipies. The work has potential. However, some points listed below need to be improved.

Abstract: I suggest add more numerical values to the abstract.

Introduction: the introduction section is too short. Please add more previously studies about Oudemansiella raphanipies.

Section 3.2.1: In my opinion, this subsection must be part of materials and method section.

Section 3.2.3 – Page 11: please better discuss why alkali concentration promotes an increase on extraction rate. Please also better discuss why “the influence of ratio of material to liquid on extraction rate rose more rapidly than that of extraction temperature and time”.

Minor editing of English language required

Author Response

To reviewer-2:

Response: Thank you for your efficient work in reviewing our manuscript. We are sorry for our carelessness.

(1) To ensure that the number of words does not exceed 200, more numerical values have been added to the abstract, which were shown in Page 1.

(2) More previously studies about Oudemansiella raphanipies have been added, which were shown in Page 2 and Page 22.

(3) The Section 3.2.1 has been moved to the part of materials and method section, which was shown in Page 5.

(4) The Section 3.2.3 has been changed into 3.2.2, more discuss about the influence of different extraction parameter has been added, the effect of alkali concentration was discussed in Page 8; which was shown in Page 11-12.

Reviewer 3 Report

The study entitled <Extraction, preliminary characterization and antioxidant activities of polysaccharides from Oudemansiella raphanipie> refers to the particular polysaccharides from a medicinal mushroom, mentioned in (traditional) medicine in India, Australia, Japan, Korea, Thailand, and China, from technological, analytical and biological activity point of views. 

As a general observation, both in vitro and in vivo studies lack the comparison with the usual, common form of the total polysaccharides from Oudemansiella raphanipie, in the general consumption mode - the mushroom powder as such.

 Point by point:  

- the technological optimization study is very meticulous, by studying the parameters of the extraction of polysaccharides from Oudemansiella raphanipie in their interrelation and combination mode;

 - the polysaccharides fractions were properly isolated, physical-chemical characterized, and antioxidant - anti-inflammatory activity studied, by both, in vitro (using several chemical models) and in vivo approaches (on several groups of mice feed with normal and high cholesterol diet at three test concentrations /50-100-200 mg/Kg body);

- the methods are adequate to the purpose of the each particular study; 

- the results are properly presented and explained in the Tables and Figures;

- the comments are consistent with the results of the investigations;  

- the conclusions are consistent with the results of the study, but need to be completed;

- thus, the strong point of the study consists in the data processing of the technological studies leading to the best conclusions on parameters study and polysaccharides extraction optimization from Oudemansiella raphanipie (pH, extraction ratio, temperature, and time);

- the weak point of the study comes from the lack of comparison with the most common form of use, the mushroom powder as such, in fact the most important conclusion for the readers;

- Finally, the authors should add some recommendations of use and regarding the significance of the results in the study, kipping in mind the normal (digestive) route of the plant polysaccharides from human diet.

Author Response

-Reviewer 3

Response: Thank you for your efficient work in reviewing our manuscript. We are sorry for our carelessness.

(1) In deed, both in vitro and in vivo studies should compare with the usual, common form of the total polysaccharides from Oudemansiella raphanipie, because the funds were finite, so the emphasis was put on the comparing the purified Oudemansiella raphanipie polysaccharides, thank you for your recommendations and opinions.

(2)- the polysaccharides fractions were properly isolated, physical-chemical characterized, and antioxidant - anti-inflammatory activity studied, by both, in vitro (using several chemical models) and in vivo approaches (on several groups of mice feed with normal and high cholesterol diet at three test concentrations /50-100-200 mg/Kg body), thank you for your recommendations and opinions.

(3)- the methods are adequate to the purpose of each particular study; thank you for your recommendations and opinions.

(4)- the results are properly presented and explained in the Tables and Figures; thank you for your recommendations and opinions.

(5)- the comments are consistent with the results of the investigations, thank you for your recommendations and opinions.

(6)- the conclusions were added, which were shown in Page 21, thank you for your recommendations and opinions.

(7)-the conclusions on parameters study and polysaccharides extraction optimizationfrom Oudemansiella raphanipie (pH, extraction ratio, temperature, and time) were added, which were shown in Page 21, thank you for your recommendations and opinions.

(8)- the weak point of the study comes from the lack of comparison with the most common form of use, the mushroom powder as such, in fact the most important conclusion for the readers, because the funds were finite, so the emphasis was put on the comparing the purified Oudemansiella raphanipie polysaccharides, thank you for your recommendations and opinions.

(9)- Some recommendations of use and regarding the significance of the results in the study have been added, which were shown in Page 21, further studies should be undertaken to research the normal (digestive) route of the polysaccharides from human diet.

Thank you again for the suggestions and work from all the editors and reviewers of Polymers.

Best wishes

Sincerely yours,

Junqiang Qiu

Top of Form

Bottom of Form
